# Tracing Viral Transmission and Evolution of Bovine Leukemia Virus through Long Read Oxford Nanopore Sequencing of the Proviral Genome

**DOI:** 10.3390/pathogens10091191

**Published:** 2021-09-14

**Authors:** Laura A. Pavliscak, Jayaveeramuthu Nirmala, Vikash K. Singh, Kelly R. B. Sporer, Tasia M. Taxis, Pawan Kumar, Sagar M. Goyal, Sunil Kumar Mor, Declan C. Schroeder, Scott J. Wells, Casey J. Droscha

**Affiliations:** 1CentralStar Cooperative, Lansing, MI 48910, USA; Laura.Pavliscak@mycentralstar.com (L.A.P.); kelly.sporer@mycentralstar.com (K.R.B.S.); 2Veterinary Diagnostic Laboratory, College of Veterinary Medicine, University of Minnesota, St. Paul, MN 55108, USA; jnirmala@umn.edu (J.N.); vsingh@umn.edu (V.K.S.); goyal001@umn.edu (S.M.G.); kumars@umn.edu (S.K.M.); 3Department of Veterinary Population Medicine, University of Minnesota, St. Paul, MN 55108, USA; pkumar@umn.edu (P.K.); wells023@umn.edu (S.J.W.); 4Department of Large Animal Clinical Science, College of Veterinary Medicine, Michigan State University, East Lansing, MI 48824, USA; taxistas@msu.edu; 5School of Biological Sciences, University of Reading, Reading RG6 6AS, UK

**Keywords:** bovine leukemia virus, retroviral evolution, proviral load, Oxford Nanopore Sequencing, phylogenetics, targeted sequencing

## Abstract

Bovine leukemia virus (BLV) causes Enzootic Bovine Leukosis (EBL), a persistent life-long disease resulting in immune dysfunction and shortened lifespan in infected cattle, severely impacting the profitability of the US dairy industry. Our group has found that 94% of dairy farms in the United States are infected with BLV with an average in-herd prevalence of 46%. This is partly due to the lack of clinical presentation during the early stages of primary infection and the elusive nature of BLV transmission. This study sought to validate a near-complete genomic sequencing approach for reliability and accuracy before determining its efficacy in characterizing the sequence identity of BLV proviral genomes collected from a pilot study made up of 14 animals from one commercial dairy herd. These BLV-infected animals were comprised of seven adult dam/daughter pairs that tested positive by ELISA and qPCR. The results demonstrate sequence identity or divergence of the BLV genome from the same samples tested in two independent laboratories, suggesting both vertical and horizontal transmission in this dairy herd. This study supports the use of Oxford Nanopore sequencing for the identification of viral SNPs that can be used for retrospective genetic contact tracing of BLV transmission.

## 1. Introduction

Enzootic bovine leukosis (EBL) is a disease caused by the retrovirus bovine leukemia virus (BVL) and is a major economic detriment to the U.S. dairy industry. Most infected cattle remain asymptomatic but approximately 30% develop persistent lymphocytosis and elevated number of B-cells, while 5% of them develop B-cell leukemia/lymphosarcoma [1]. BLV pathogenesis compromises the host’s immune system, leaving the cows susceptible to several other opportunistic pathogens, all of which might serve as a potential disease reservoir within the herd and ultimately shorten the lifespan and productivity of the cows.

Many European countries have successfully eradicated BLV using bulk tank surveillance. However, our group has found that 46% of dairy cows in 94% of dairy farms in the United States are infected with BLV [2], which is consistent with the reports from USDA APHIS (https://www.aphis.usda.gov/animal_health/nahms/dairy/downloads/dairy07/Dairy07_is_BLV.pdf accessed on 6 January 2021). In addition to affecting the longevity and overall health of the cows, BLV has also been found to affect milk production, which accounts to a loss of approximately 380 USD per cow per year [3]. Detection of BLV DNA in human blood or tumor samples has been linked to potential human health concerns, including breast cancer [4,5].

BLV primarily infects B-cell lymphocytes, and following proviral integration, is propagated through mitosis of immature B-cell progenitors. Transmission of BLV is thought to occur through the introduction of an infected lymphocyte to the bloodstream of the uninfected animal through milk/colostrum and blood-borne routes, including biting flies and iatrogenic means such as shared needles and rectal palpation sleeves [6]. However, the use of best management practices has proven to reduce within-herd prevalence of BLV significantly [7]. Vertical transmission, in utero, during parturition, or postnatally via colostrum administration, has also been found to contribute to 4–18% of new BLV infections [8,9]. Feeding BLV-infected unpasteurized colostrum [10] and contact with bodily fluids in the birth canal during the delivery [11] have all been found as potential routes of BLV transmission in neonatal calves. Lastly, viral transmission through the placenta could result in a positive calf at birth [11].

The proviral load of infected animals may be directly correlated with the animal’s ability to infect others, as with other retroviral diseases such as HIV [12]. In fact, a multi-herd BLV control field trial [13] showed a reduction in the within-herd incidence of BLV infection associated with the removal of high proviral load cattle from the herd. Antibody detection via Agar Gel Immunodiffusion (AGID) and Enzyme-linked Immunosorbent Assay (ELISA) are currently accepted methods used to diagnose cows but are poorly correlated to an animal’s proviral load.

Detection of BLV proviral DNA can be accomplished through endpoint polymerase chain reaction (PCR) with varying levels of specificity and sensitivity, depending on the design of primer sequences. Nested end-point PCRs are commonly used as a highly sensitive method of to detect BLV proviral DNA, despite their poor specificity [12,14,15,16]. Quantitative PCR (qPCR) allows for the relative and simultaneous quantification of provirus and host DNA with a high degree of specificity and sensitivity when used with TaqMan^®^ fluorophores. Our group developed the SS1 BLV proviral load qPCR multiplex assay (CentralStar Cooperative, Lansing, MI, USA) consisting of three TaqMan™ primer/probe assays that detect the BLV *pol* gene, bovine *β-Actin*, and an internal amplification control [17,18,19]. Through qPCR calibration using digital droplet PCR (ddPCR) normalized DNA standards, the SS1 assay has 100% specificity and sensitivity down to a limit of detection of 10 BLV copies. We therefore can precisely compare the relative infectiousness between animals over time, using the SS1 BLV proviral load qPCR multiplex assay.

To date, genetic diversity of the BLV proviral DNA has been characterized mostly based on sequencing the envelope gene (*env*) either by Sanger or short-read next-generation sequencing (NGS, Illumina) methods. Ten different genotypes have been identified so far based on the variations within the *env* gene and whole genome analysis via tiled PCR based NGS sequencing [20,21]. Previous studies agree that BLV genotypes associate with the global geographical origin [21,22]. However, it is precarious to draw on international BLV genetic relationships based on short and conserved genomic regions, like the *env* gene. Sanger sequencing and short-read NGS have established important geographical and molecular relationships between different viral variants, but the utility of these approaches for elucidating the genetic structure and high-resolution genetic relationships of BLV variants within confined populations is limited.

Single-molecule long-read sequencing methods, such as Oxford Nanopore Technologies (ONT), have enabled the sequencing of large genetic elements spanning across repetitive regions and eliminate common sequencing issues encountered with other technologies, such as high G/C content and palindromic regions. The real-time nature of ONT significantly reduces the long turnaround times involved in Illumina sequencing and is considerably less expensive [23]. This methodology would also enable accurate and timely identification of novel genotypes, which can be critical to effectively identify emerging pathogens. This pilot study aimed to develop a molecular sequencing tool that can be used to accurately identify new BLV variants and discern routes of transmission while simultaneously assessing the origin, evolution, and genetic diversity of BLV within herds.

## 2. Results

### 2.1. ONT Sequencing of Long-Range PCR Amplicons from TOPO ^®^ Vectors Containing Bacterial Inserts

Prior to drawing conclusions on genetic variability among BLV sequences from related animals, it was important to determine the feasibility and accuracy of the experimental workflow and bioinformatics pipeline. PCR amplicons derived from mastitis-causing microorganisms, ligated into TOPO ^®^ plasmids, and verified by Sanger sequencing, were used to evaluate the fidelity of the developed workflow. Approximately 4 kb PCR products were generated using primers with 5′ overhangs containing the BLV_CS forward and reverse primer sequences (Table 1, Appendix A). Secondary TOPO ^®^ PCR products produced from BLV_CS primers were sequenced on the ONT GridION. The resulting sequences were compared to their corresponding Sanger sequence-verified vectors to confirm sequencing accuracy. Both sets of vectors with their associated inserts, either a 16s rDNA sequence from *Corynebacterium bovis* or Thioredoxin (*TrxA*) from *Mycoplasma bovis*, were found to be 100% identical (Figure 1) with an average 2900 × coverage (Appendix A).

### 2.2. ONT Sequencing of BLV Whole Genome Sequences

Blood samples were collected on a single date from sixteen BLV ELISA-positive adult dam and daughter Holstein pairs from a commercial dairy farm. BLV proviral load was determined by the SS1 BLV proviral qPCR multiplex assay (CentralStar Cooperative, Lansing, MI, USA; Table 2) [14]. Following organic phase separation and a modified high molecular weight (HMW) DNA extraction [24], 15 DNA samples were split into two aliquots for PCR amplification of the BLV proviral genome. Approximately 7.5 kb PCR products were amplified using two different sets of primer pairs developed by two independent laboratories (Figure 2). The negative BLV SS1 sample (15.1) was consequently removed from the study. Electrophoretograms of the amplicons generated from the two laboratories were compared to evaluate the PCR efficacy of two different primer sets (Figure 3). The number of trimmed reads per corresponding sequencing adaptor and the coverage of each amplicon were evaluated to determine amplicon quality and infer the accuracy of subsequent analyses. The differences seen with a cross-laboratory comparison of bands could be due to read depth variability caused by amplicon quality which can confound phylogenetic interpretation. The minimal number of reads generated from the NTC (Non-Template Control) are likely due to barcode bleed-through resulting from the use of single barcodes as opposed to dual barcodes [25].

### 2.3. Phylogenetic Analysis of BLV Whole Genome Amplicons

Phylogenetic analysis of the ONT consensus sequences confirmed the fidelity of the independent PCR approaches and served to assess the genetic diversity of the BLV found within a herd. Ten out of the 14 pairs, dams 1–7 and daughters 5–8 of technical replicates from the two laboratories matched on taxa location on the tree, solidifying the accuracy of both PCR techniques along with the sequencing of generated amplicons (Figure 4). The outlying samples, daughters 1, 2 and 3_UMN, were concluded as resulting from poor coverage of those amplicons (Appendix A).

When analyzing samples based on familial relationships, the distance between proviral sequences from dams and daughters varies. Pair five, for example, shows sequence identity matching based on clade location. This phylogenetic analysis also displayed viral divergence of BLV strains in one generation. In dam/daughter pair two, the strain found in the daughter branches off of the node shared by the dam, representing viral evolution. The fidelity of the sequencing was confirmed via the bacterial TOPO ^®^ plasmids, confirming viral evolution rather than sequencing error. These results indicate that this tool is capable of discerning the route of infection while capturing viral diversity and evolution with herds over time.

Historically, the majority of BLV phylogenetic analyses is based solely on the *env* gene sequences alone (18–20) [20,21,22]. Analysis of the *env* gene sequences extracted from the ONT reads within this study (Figure 5) identified only six different nucleotide substitutions which were synonymous at the amino acid level. (Appendix A). Env polyprotein was completely conserved between the animals in the herd studied, which would significantly limit the understanding of genetic variability between the animals in this herd if only this BLV gene was sequenced.

### 2.4. Amino Acid Substitution among BLV Positive Animals

To further compare the dam and daughter pair relationship, we performed a comparative amino acid analysis of six well defined proteins that might play a role in disease dynamics (Table 3. Only the sequences generated from the BLV_CS primers were used due to the variability in read depth and coverage seen in the BLV_UMN amplicons. Amino acid profiles from five dams, 71%, (1, 3, 4, 6 and 7) diverged from their daughters. On the other hand, only 2/7 (29%) of the dams and daughters shared identical amino acid profiles. The highest number of amino acid changes was observed between dam/ daughter pair 1 (4 AA substitutions), followed by the pair 7 (4 AA substitutions). At the individual protein level, no functional SNPs could be observed in the R3 protein. Gag Pro Pol protein was the most polymorphic with the highest number of functional SNPs (4 AA substitutions) followed by the *env* polyprotein (3 AA substitutions). Despite the variation observed with the *env* polyprotein, it is very interesting to note that the env protein was absolutely conserved among the animals investigated from this herd. These results clearly show a significant level of functional variations/modulation between the proteins and the dam daughter pairs investigated, even within a small set of animals investigated from a single herd.

## 3. Discussion

This study successfully utilized a bioinformatics pipeline, which included Longshot, to call single nucleotide variants with high accuracy using whole-genome ONT data and assemble near complete BLV proviral consensus genomes from field-collected samples. The use of vectors with inserts verified by the Sanger sequence was able to support the integrity of the sequencing reads developed by the ONT workflow. The BLV genome from the DNA of whole blood samples were independently amplified twice, in two different laboratories, following proviral load testing. Phylogenetic analysis of all resulting consensus genomes highlighted the accuracy of the independent amplification of the proviral genomes due to the location of the duplicated samples. Using this study data, we also defined differing transmission routes of BLV within the herd, demonstrating the value of this tool to characterize primary transmission pathways in tested populations. The utilization of this tool also enables further research into the tracing of genetic elements, such as the proviral genomes of retroviruses infecting cattle.

### 3.1. Fidelity of Approach

Development of long-read BLV genome PCR occurred independently between the two participating laboratories with similar objectives, to assess BLV viral transmission. Therefore, primer design criteria, including BLV reference sequences, and PCR condition considerations were individually optimized respective to the polymerase used. This approach allowed for the evaluation of the robustness of the sequencing workflow and downstream interpretation. The efficiency of PCR reactions differed resulting in variation seen in Figure 3, corresponding to coverage achieved. However, the sequencing workflow was able to overcome PCR inefficiencies and generate BLV proviral genome sequences.

Due to the fact that a novel library preparation and bioinformatics pipeline were used, the fidelity of the generated reads needed to be evaluated. Oxford Nanopore Technology sequencing has been previously criticized for its high error rates [26,27] so it was crucial to confirm the accuracy of the resulting sequences. When comparing the sequences of the TOPO ^®^ vectors with the known bacterial inserts generated by both Sanger sequencing and ONT, 100% sequence identity was found (Figure 1). Along with proven identical sequences, the read depth generated by our developed workflow helps overcome the high error rate, as seen in other studies as well [28]. This is a significant obstacle to have overcome and allows for greater insights into nucleotide substitution of large cis-acting genetic elements, such as viral genomes.

A stepwise validation was used as a quality control check at each point along the pipeline. A time intensive HMW DNA extraction was used to ensure high-quality DNA and was confirmed via Nanodrop quantification and gel electrophoresis (Data not shown). Success of the BLV PCR was evaluated with gel electrophoresis and the use of the bacterial amplicons as positive sequencing controls. Following sequencing, reads were trimmed and filtered with our developed bioinformatics protocol. After visualizing the generated genomes on a phylogenetic tree and sequencing coverage plots, unreliable samples with poor PCR product quality and low coverage were removed from future analysis to increase the confidence of generated results (Appendix A).

### 3.2. Reflection of Findings and Implications for the BLV Field

Despite the high estimated prevalence of BLV within the United States and the asymptomatic nature of the retrovirus infection, disease control is challenging for dairy cattle producers. Without availability of effective treatment or vaccination programs, effective herd management practices to break the chain of transmission is critical. Various transmission routes of BLV have been reported [19,20,21,22,23], but it has been challenging to find direct evidence to prove vertical transmission of the virus. We sought to develop a tool to accurately analyze the majority of the proviral BLV genome to aid in the detection of transmission among United States dairy farms. This method is likely to be important for comparisons among a larger scale population of samples from varying geographical locations. To test this approach, we selected dam and daughter pairs that were both BLV-positive to assess the genetic identity of the proviral genomes.

On a local level, this phylogenetic analysis tool can help dairy farmers identify patterns of BLV transmission in their herds. Due to the assortment of viral variants, dam-daughter pairs with identical genetic sequences were concluded to likely be a result of vertical transmission (Figure 4). However, it could be possible that another animal passed on that exact sequence to both dam and daughter. In this study herd, only milk replacer is used for dairy replacement calves and not colostrum. This increased the likelihood of vertical transmission from prenatal or perinatal infection and not transmission post-calving. Contrastingly, daughter 6, for example, was likely infected horizontally due to the greater genetic distance between her and her mother’s strain (Figure 4). Increased rates of vertical transmission could be decreased with do not breed orders implemented on BLV positive dams or screening practices established before artificial insemination to ensure all bred cows are BLV negative. Separately, increased rates of horizontal transmission could be reduced with improved bloodborne biosecurity and better isolation practices used for BLV-positive cattle.

Globally, BLV sequences have been mostly categorized into 10 different genotypes based on the analysis of the *env* gene. The use of small proviral fragments heightens the difficulty in drawing phylogenetic and evolutionary conclusions. In our study, an amino acid alignment of just the *env* gene sequences resulted in 100% identical peptide sequences among all samples (Appendix A). The utilization of only the *env* gene would not have provided the level of resolution required to discern different routes of transmission gained when using whole-genome sequence data. The use of entire proviral genomes would aid in the comparison of international strains, compared to conclusions drawn from analysis of a singular gene. Amino acid analysis of these genomes could increase awareness of BLV pathogenesis and virulence differences among genotypes. Functional changes resulting from mutations in Tax, Rex, and G4-along with multiple other proteins have been previously studied, but do not align with the mutations detected in our sample set [29,30,31,32,33]. More research is needed to see if the amino acid changes listed in Table 3 impact BLV pathogenicity and transmissibility.

### 3.3. Application

We have developed a tool to accurately determine full-length BLV proviral genome sequences. Within the field of BLV research in cattle, Oxford Nanopore sequencing of whole targeted BLV whole-genome amplicons will aid in tracing spatial and temporal modeling of infections within different herds and elucidate how cattle movement and farm management practices affect transmission. This approach can also be used to help resolve genetic relationships of dynamic viral populations of other pathogens such as SARS-CoV-2. Li et al. used a similar ONT workflow with SARS-CoV-2 samples to identify various sources and transmission patterns in China [34]. The developed workflow could also be a valuable tool for resolving transmission patterns of Human T Lymphotropic Virus type 1 (HTLV-1) in the central regions of Australia where disease prevalence has been estimated at 33% within indigenous populations [35] as this virus most similarity related to BLV. Furthermore, this tool could be used to resolve structural variation within several genomic loci at once to extend GWAS-based approaches that identify highly associated SNPs with a given phenotype within humans and livestock [36,37].

### 3.4. Limitations to This Pilot Study

The sample set used in this study is relatively small and does not represent all dairy herds within the United States. The data collected on selected animals only displays one snapshot within their lifetimes. A longitudinal study, including BLV-positive neonate samples is needed to evaluate direct and temporally relevant transmission patterns. This sampling strategy could also identify changes in the genetic diversity of the viruses within a herd and draw more concrete conclusions on the routes of transmission.

The proviral DNA was the main sample source used, not RNA. Future research could analyze BLV RNA transcripts to analyze strain functionality differences. The proviral DNA was extracted using a time-intensive high molecular weight protocol. The efficacy of this tool with extracts derived from other extraction protocols, such as column-based, needs to be evaluated in the future. The use of less time-consuming DNA extractions would lead to the better universal application of this tool. However, the use of the HMW protocol allows for future direct DNA sequencing, without PCR amplification, to potentially evaluate the differential states of DNA methylation of the BLV proviral genome, its transcription and resulting disease dynamics [38].

## 4. Materials and Methods

### 4.1. Sample Collection

Whole blood was collected in K2 EDTA Vacutainer^®^ tubes (BD Vacutainer, Franklin Lakes, NJ, USA) via coccygeal vein of eight BLV ELISA-positive dam-daughter pairs of Holstein cows on a single date (Table 1). Samples were collected from these adult cattle ranging from 25 to 91 months of age. Procedures for this study were reviewed and approved by the Institutional Animal Care and Use Committee (A-3955-01, PROTO201900271) at Michigan State University.

### 4.2. BLV Antibodies

Individual DHI milk samples were tested on the same date, with the Leukosis Serum × 2 Ab test (IDEXX, Westbrook, ME, USA) to identify the BLV serostatus of the animals at the time of bleeding. In short, milk samples were diluted in sample buffer and pipetted into 96-well plates coated with BLV-GP51 antigen per the manufacturer’s instructions. Horseradish peroxidase-labeled bovine anti-immunoglobulin was added followed by incubation at room temperature for 5 min. Plates were washed after each incubation and before adding the enzyme-substrate. Reaction times were standardized using color development of positive controls and stopped by adding 0.5 mL Sulfoamino oxidanide (H_2_NO_4_S). Results were reported as corrected 450 nm optical density (OD) measurements with a corrected OD > 0.5 being considered antibody positive.

### 4.3. Proviral Load Diagnosis of BLV-Infected Animals

Qiagen DNeasy blood and tissue kit (Qiagen, Hilden, Germany) was used to extract genomic DNA from whole blood collected from previously ELISA-positive cows one week following milk ELISA screening. The SS1 qPCR assay, developed by CentralStar Cooperative Inc., is a multiplex probe-based quantitative PCR assay that targets the BLV proviral polymerase gene, bovine *β-Actin* gene, and an internal amplification spike-in control ultramer to quantify proviral load. Briefly, 3 µL of extracted DNA, 12.5 µL of 2 × PrimeTime Gene Expression Master Mix (ThermoFisher, Austin, TX, USA), 1.25 µL of a 20 × primer mix, 1 µL of internal spike-in control (10,000 copies/µL), and 7.25 µL of DNA-free water were combined for each qPCR reaction. All SS1 qPCR was performed on Applied Biosystems 7500 Fast Real-Time PCR system (FAST Real-Time PCR, Foster City, IA, USA) with qPCR conditions as follows: 95 °C for 10 min, 40 × (95 °C for 15 sec, 60 °C for 1 min). BLV and bovine *β-Actin* (a measure of host DNA) copy numbers were derived using a standard curve consisting of linearized plasmids containing respective target sequences previously quantified and normalized by digital droplet PCR. Amplification efficiency and manual thresholds were established from initial qPCR machine calibration. Proviral Load was calculated and expressed as the ratio between proviral BLV copies and bovine *β-Actin* copies.

### 4.4. High Molecular Weight DNA Isolation

Frozen aliquots containing 500 µL of whole blood were thawed to room temperature and added to 1 mL of 1 × ChemCruz red cell lysis buffer (RBC: Santa Cruz Biotechnology, Dallas, TX, USA), mixed via inverting, and incubated at room temperature on a rotisserie tube rotator for 10 min. Samples were centrifuged at 4700× *g* for 5 min and the supernatant was removed. This was repeated once more with 500 µL of RBC lysis buffer. White blood cell pellets were resuspended in 1.5 mL of modified mammalian cell lysis buffer [26] and incubated at 37 °C for 1 h prior to adding 100 µg/mL of Proteinase K (Invitrogen, Waltham, MA, USA) and mixing with a wide bore pipette. The samples were incubated at 55 °C for an hour with occasional agitation. 500 µL of buffer-saturated phenol (Invitrogen, Waltham, MA, USA) was added at room temperature and mixed on the tube rotator for 20 min. Organic phase separation was accomplished via centrifugation at 5000× *g* for 15 min. The aqueous phase was transferred to a fresh tube and 500 µL of phenol-chloroform-isoamyl alcohol (Acros Organics, Fair Lawn, NJ, USA) was added. After an additional phase separation step, 0.266 volumes of 7.5 mM ammonium acetate were added to the aqueous phase. Following a 5 min incubation at room temperature, 100% ethanol was added followed by thorough mixing. The precipitate was collected via centrifugation at 8000× *g* for 8 min at room temperature. Pellets were then dehydrated in a succession of 90% and 70% ethanol washes followed by centrifugation at 8000× *g* for 8 min. DNA pellets were air-dried, eluted in 50 µL of 7.5 pH Tris-EDTA (10 mM Tris,0.1 mM EDTA), and incubated overnight at 4 °C. DNA was quantified using a Nanodrop spectrophotometer and stored at −20 °C.

### 4.5. BLV Whole Genome PCR Amplification

#### 4.5.1. CentralStar BLV Whole Genome PCR

To summarize, 1 µL of extracted HMW-DNA, 10 µL 5 × LongAmp Taq reaction buffer (New England Biosystems, Ipswich, MA, UK), 1.5 µL 10 mM dNTPs, 5 µL 0.5 uM forward and reverse primers (Table 1: BLV_CS), 2 µL LongAmp Taq DNA Polymerase, and 22.5 µL of DNA-free water were combined for each PCR reaction. PCR was run on an Applied Biosystems 2720 Thermal Cycler (Applied Biosystems, Waltham, MA, USA) with conditions as follows: 94 °C for 30 s, 29 × (94 °C for 20 s, 64 °C for 25 s, 65 °C for 6 min 45 s), 65 °C for 10 min. Amplicons were run out on a 1% agarose gel at 100V for an hour to confirm amplification before library preparation.

#### 4.5.2. University of Minnesota Veterinary Diagnostic Laboratory (MVDL) BLV Whole Genome PCR

High molecular weight DNA extracted (as described above) was used as a template to amplify a 7604 bp amplicon corresponding to the region, 550–8154 bp of BLV whole genome. A 50 μL PCR reaction mixture containing 10 μL of 5 × Primestar GXL buffer, 4 μL of dNTPs (2.5 mM each), 1.5 μL (15 pm) of each primer (Table 1: BLV_UMN), 1 μL of GXL Primestar Polymerase (Takara Bio USA Inc., Mountain view, CA, USA), 28 μL of nuclease-free water (Ambion), along with 4 μL of the template was prepared. PCR mixture was initially incubated for 10 s at 98 °C for denaturation, followed by 30 cycles of 60 °C for 15 s and 68°C for 2.5 min in a MiniAmp Plus Thermal cycler (Applied Biosystems, Waltham, MA, USA)

### 4.6. Use of Vectors Containing Bacterial Inserts as Positive Sequencing Controls

To measure the fidelity of the sequencing approach, previously generated linearized TOPO2.1 ^®^ (Thermo Fisher Scientific, Austin, TX, USA) vectors containing bacterial genes (*Mycoplasma bovis -TrxA, Corynebacterium bovis-16 s*) were used as PCR templates. Briefly, the TOPO2.1 ^®^ vector containing 123 bp insertion of *M. bovis* thioredoxin DNA was linearized using the *Sca* I restriction enzyme. Topo-BLV primers (Table 1) were used to target the 5′ and 3′ ends of the linearized vector producing a 3945 bp amplicon. The linearized vector containing the 143 bp insert of *C. bovis* 16s rDNA resulted in a 3967 bp amplicon (Appendix A). BLV_CS primers were used to amplify the primary TOPO ^®^ amplicons prior to Oxford Nanopore sequencing.

### 4.7. Bead Clean Up and Library Preparation for Oxford Nanopores Sequencing

One μL of the BLV or positive control PCR amplicon/s were electrophoretically separated and visualized on a genomic DNA tape screen using a 4200 TapeStation (Agilent Technologies, Santa Clara, CA, USA) for size and integrity. All amplicons, irrespective of size, served as templates for ONT GridION sequencing. The remaining BLV and positive control PCR amplicons were subjected to a 2 × bead clean-up using Sera-Mag Select beads (GE HealthCare Life Sciences, Chicago, IL, USA) and eluted in a final volume of 12 μL of nuclease-free water. The purified PCR product was finally quantified using the Qubit 1X dsDNA HS assay kit (Thermo Fisher Scientific, Waltham, MA, USA) using a Qubit 4 fluorometer (Thermo Fisher Scientific, Waltham, MA, USA) and then used for library preparation.

Library preparation was performed using the Rapid Barcoding Sequencing Kit (SQK-RBK004) (Oxford Nanopore Technologies, New York, NY, USA). Briefly, 7.5 μL of the purified BLV amplicons were mixed with 2.5 μL of a fragmentation mix barcode individually (1 barcode for each sample; 12 samples were barcoded and used on a single flow cell, wherever possible), incubated at 30 °C for 1 min and 80 °C for 1 min, followed by rapid cooling on ice. One μL each of the 12 barcoded libraries were pooled, mixed gently with 1μL of RAP (Rapid Adapter), and incubated for 5 min at room temperature. This final library pool (13 μL) was combined with 34 μL of sequencing buffer, 25.5 μL of loading buffer, 2.5 μL nuclease-free water (total of 75 μL) and loaded onto a primed Flow Cell R9.4 (FLO-MIN106D) on a GridION device and run with the SQK-RBK004_plus_Basecaller script of MinKNOW1.5.12 software. The run was stopped after 4 h, and the flow cell was washed with a Wash Kit (EXP-WSH002) (Oxford Nanopore Technologies, New York, NY, USA) according to the manufacturer’s recommendation and stored at 4 °C for later use.

### 4.8. Bioinformatic Analysis of Oxford Nanopore Sequencing and Consensus Sequence Generation

The sequenced reads were base called using *Guppy* (4.0.14) and demultiplexed and filtered with *qcat* v1.1.0 (ONT, https://github.com/nanoporetech/qcat accessed on 2 June 2021) with parameters *--detect-middle --min-read-length 250 --trim*. Reads were aligned to the BLV reference genome (AP019598.1) using *minimap2* v2.17 [39] and primer sequences were trimmed from the termini of read alignments using *iVAR* v1.3.1 [40] with parameter *-p trim -q 1 -m 20 -s 4 -e*. Consensus-level variant candidates were identified using *Medaka* v1.2.2 (https://github.com/nanoporetech/medaka accessed on 2 June 2021) using default parameter and evaluated by *LongShot* v0.4.1 [41] with parameter *-P 0 -F -A --no_haps* before filtering with bcftools. The final consensus was generated from a filtered VCF file and depth of coverage extracted from mapping file with samtools depth with parameter *-a -d 0*.

### 4.9. Phylogenetic and Amino Acid Analysis of BLV Whole-Genome Consensus Sequences

A comparative phylogenetic analysis was performed to investigate the genetic relatedness of BLV genomes within related Holstein dams and adult daughters within a Michigan dairy herd All the sequences were aligned with high accuracy and high throughput using MUSCLE (MUltiple Sequence Comparison by Log- Expectation) with 64 iterations [42] (National Center for Biotechnology Information, Bethesda, Maryland, USA). Phylogenetic trees were constructed using maximum-likelihood (ML) methods [43] for each segment with 1000-bootstrap replications and GTR Gamma nucleotide substitution in RAXML v7.6.8, utilizing the resources available at the Minnesota SuperComputing Institute. Trees were rooted to the midpoint, with the increasing order of nodes. Tree topology was supported by >70 bootstrap values. Bootstrap values below 70 were manually removed from the trees using Adobe Illustrator CC 2018 (version 22.0.1 accessed on 7 March 2021). Trees were visualized and depicted using FigTree (version 1.4.3 accessed on 7 March 2021). BLV nucleotide sequences, translated to amino acids using Geneious Prime11.0.6 + 10, were used to identify nonsynonymous mutations, if any, among the major proteins of BLV using AP019598.1 as a reference genome. 

## Figures and Tables

**Figure 1 pathogens-10-01191-f001:**
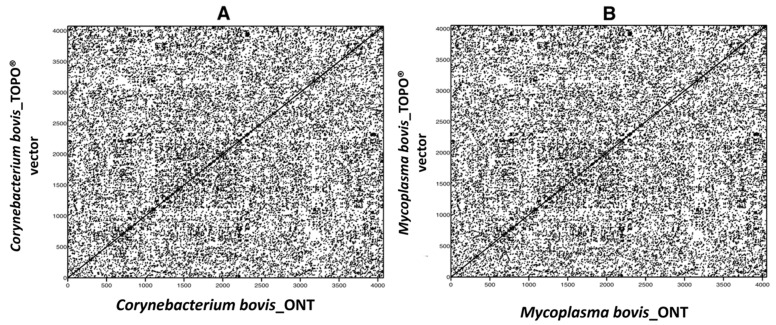
DNA dot plot matrix showing 100% similarity between sequences derived from Oxford Nanopore Technology (ONT) and Scheme 16s rDNA (**A**) or Thioredoxin gene (**B**) from two different mastitis-causing pathogens. (**A**): *Corynebacterium bovis* and (**B**): *Mycoplasma bovis*.

**Figure 2 pathogens-10-01191-f002:**
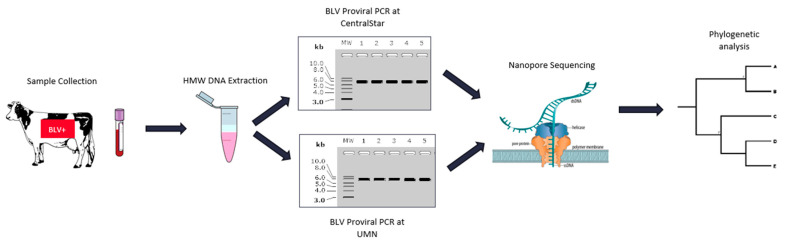
Experimental workflow for inter-laboratory technical replication of long-range BLV genome PCR amplification and Oxford Nanopore Sequencing.

**Figure 3 pathogens-10-01191-f003:**
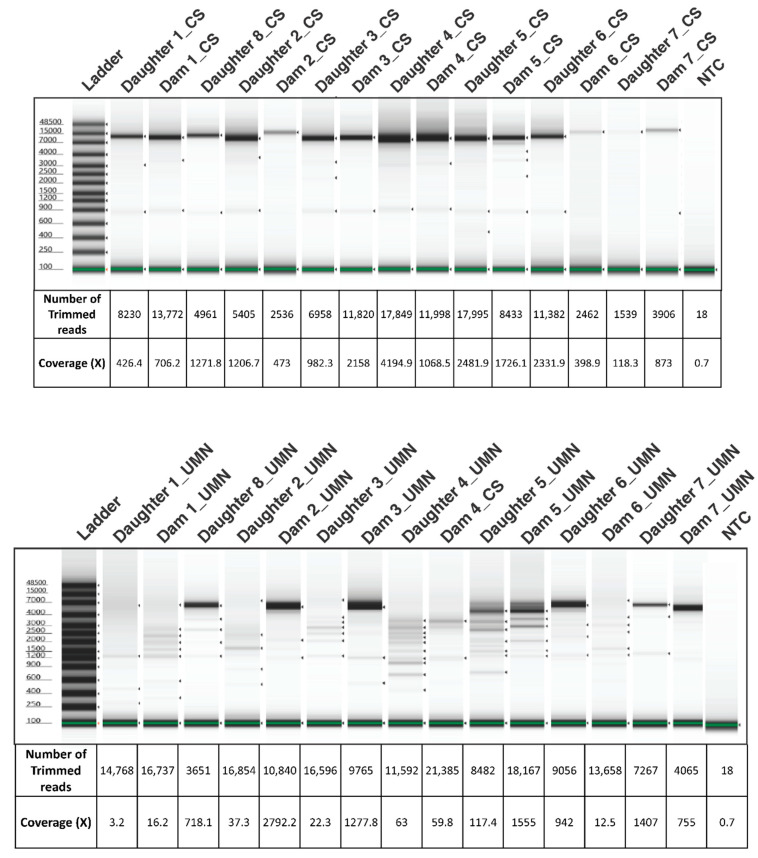
Electrophoretograms showing the amplification, size, and integrity of BLV amplicons used for Oxford Nanopore Sequencing developed by the UMN and CentralStar methods. The coverage and number of trimmed reads used for mapping to the BLV genome is provided below each lane corresponding to the sample. Samples from CentralStar and UMN laboratories are appended with the letters “CS” and “UMN” respectively following the dam or daughter pair number. NTC: Non-Template Control.

**Figure 4 pathogens-10-01191-f004:**
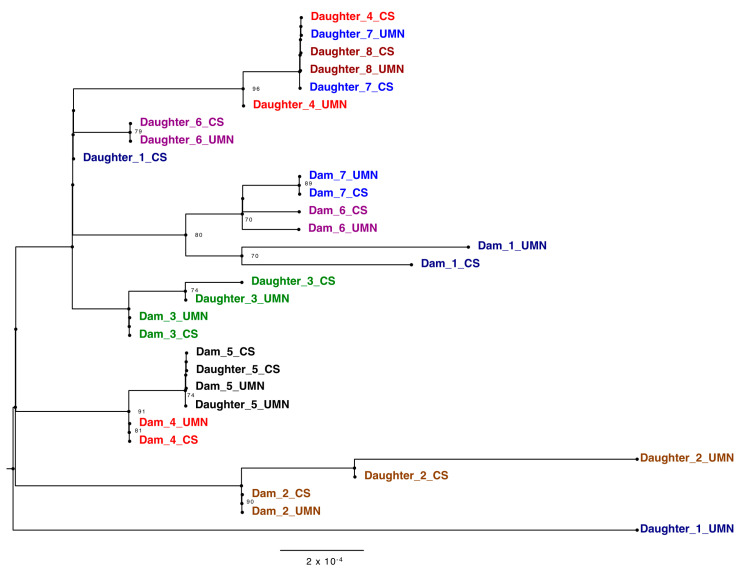
Phylogenetic tree of BLV genomes derived from amplicons generated in the CentralStar and University of Minnesota laboratories displaying vertical and horizontal transmission between the dam-daughter pairs. Trees were inferred using maximum likelihood methods and supported with bootstrap value above 70. Different pairs are shown in different colors. Samples from CentralStar and UMN laboratories are appended with the letters “CS” and “UMN” respectively following the dam or daughter pair number.

**Figure 5 pathogens-10-01191-f005:**
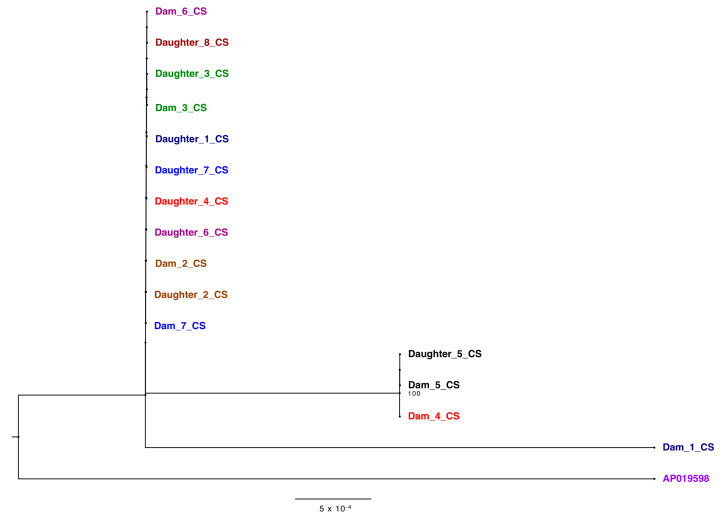
Phylogenetic tree of *env* gene derived from sequences generated from the CentralStar laboratory. Different pairs are shown in different colors. Trees were inferred using maximum likelihood methods and supported with bootstrap value above 70.

**Table 1 pathogens-10-01191-t001:** Sequences of primers used for amplifying the sequencing controls and BLV whole genomes. Sequences in bold represent the BLV primer tags that were incorporated into the plasmids for use as ONT sequencing positive controls.

*Primer*	Direction	Sequence	Length	Tm	%GC	Accession
*BLV_CS*	Forward	5′-AACCTTCTGCAAAGCGCGCAAA-3′	22	68	50	AF033818
	Reverse	5′-AAGGCGGGAGAGCCATTCATTTTC-3	24	67	50	
*BLV_UMN*	Forward	5′-ATTGATCACCCCGGAACCCTAAC-3′	23	66	52	AP019598.1
	Reverse	5′-CTCAAAAAAGGCGGGAGAGCCATTC-3′	25	68	52	
*Topo-BLV*	Forward	5′-**AACCTTCTGCAAAGCGCGCAAA** GCGGCCAACTTACTTCTGACAAC-3′	45	70.1	51.1	PCR ^®^2.1-TOPO ^®^
	Reverse	5′-**AAGGCGGGAGAGCCATTCATTT** AATAGTGTATGCGGCGACCGAGT-3′	45	69.7	51.1	
*M. bovis*	Forward	5′-GGTTAATTCTATGCCCAGCATT-3′	22	60	41	CP058514
	Reverse	5′- TTCAGCTTCAATTGCATCCAC-3′	21	60	43	
*C. bovis*	Forward	5′-GTGCTTTAGTGTGTGCGGTGG-3′	21	62	57	NZ_AENJ01000027
	Reverse	5′-CGTGTCTCAGTCCCAATGTGG-3′	21	60	57	

**Table 2 pathogens-10-01191-t002:** BLV Diagnostic results of animals enrolled in this study. BLV was detected from 7 dam-daughter pairs (1–7) by SS1 qPCR. Dam 8 was BLV negative and hence was removed from subsequent sequencing.

Pair	Relationship	Animal ID	Date of Birth	Age (MO)	Lactaion	PVl	PVl Category	ELISA OD
1	Daughter	1437	25 June 2017	39	1	0.91	High	2.799
	Dam	1182	11 June 2015	63	3	1.02	High	2.626
2	Daughter	1479	11 November 2017	34	1	0.37	Low	2.5
	Dam	1118	29 November 2014	70	4	0.71	Low	1.287
3	Daughter	1503	21 December 2017	33	1	0.39	Low	1.704
	Dam	980	6 October 2013	84	5	0.41	Low	2.266
4	Daughter	1283	10 February 2016	55	2	1.18	High	1.56
	Dam	909	27 February 2013	91	5	0.58	Low	2.559
5	Daughter	1523	2 February 2018	32	1	0.86	High	1.48
	Dam	1280	2 February 2016	56	3	1.18	High	0.705
6	Daughter	1560	22 May 2018	28	0	0.41	Low	0.893
	Dam	1303	27 March 2016	54	2	0.01	Low	0.5
7	Daughter	1513	3 January 2018	33	1	0.04	Low	0.768
	Dam	1209	15 August 2015	61	3	0.87	High	2.335
8	Daughter	1606	4 April 2018	25	0	0.32	Low	2.793
	Dam	1176	23 May 2015	64	3	N/A	Negative	2.476

**Table 3 pathogens-10-01191-t003:** Summary of significant amino acid changes observed in six different proteins encoded by the BLV genome among the dam-daughter pairs sequenced using primers and conditions developed by the CentralStar laboratory. Yellow: Conserved amino acid. Blue: Amino acid change.

Protein and Amino Acid Locations	SNP Position	Sample ID														
		Daughter 1_CS	Dam 1_CS	Daughter 8_CS	Daughter 2_CS	Dam 2_CS	Daughter 3_CS	Dam 3_CS	Daughter 4_CS	Dam 4_CS	Daughter 5_CS	Dam 5_CS	Daughter 6_CS	Dam 6_CS	Daughter 7_CS	Dam 7_CS
G4 (26-105)	71	D	N	D	D	D	D	D	D	D	D	D	D	D	D	D
GAG PRO POLY PROTEIN (31-1417)	465	W	W	W	W	W	W	W	W	W	R	R	W	W	W	W
	642	I	I	I	I	I	I	I	I	I	I	I	I	I	I	V
	1162	E	G	E	E	E	E	E	E	E	E	E	E	E	E	E
	1173	G	G	G	R	R	G	G	G	G	G	G	G	G	G	G
REX (1-56)	48	S	S	S	S	S	S	S	S	S	S	S	S	F	S	F
	140	L	R	L	L	L	L	L	L	L	L	L	L	R	L	R
R3 (1-44)	No SNPS															
EPP (1-515)	31	I	I	I	I	I	I	I	I	I	I	I	T	I	I	I
	475	H	H	H	P	P	H	H	H	P	P	P	H	H	H	H
	479	L	L	V	L	L	L	L	V	L	L	L	L	L	V	L
TAX 1-309	182	T	T	T	T	T	A	A	T	T	T	T	T	T	T	T
AS1 1-87	3	P	S	P	P	P	P	P	P	P	P	P	P	P	P	P
	31	G	G	G	E	G	G	G	G	G	G	G	G	G	G	G

## Data Availability

Data is contained within the article and Appendix A. Sequencing data to be reported and publicly available on a later date.

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
