# Peer review of "Tracing Viral Transmission and Evolution of Bovine Leukemia Virus through Long Read Oxford Nanopore Sequencing of the Proviral Genome"

_pathogens, 2021, doi:10.3390/pathogens10091191_

Round 1
Reviewer 1 Report
The manuscript entitles “Tracing viral transmission and evolution of Bovine Leukemia virus through long read Oxford Nanopore Sequencing of the proviral genome“ describes sequence diversity and identity of near complete BLV genomic sequences obtained by Oxford Nanopore Technologies (ONT). The study is of interest for the field of both BLV molecular epidemiology and general animal virology. In addition, this study represents the first application of the nanopore sequencing technology to analyze BLV sequences in cattle blood samples, which will be of interest to many other researchers that are getting into finding key viral sequences that relevant to BLV pathogenesis. However, major concern in this manuscript is that the high error rates of ONT (as authors mentioned in the manuscript, page 18, line232) are critical for retrospective genetic contact tracing of BLV because of low nucleotide substitution rate.
Major comments
1) The authors mainly relied on the plasmid-based analysis to validate ONT sequencing. The aliquot of samples used in the validation only contains cloned plasmid DNA, which is much easier to analyze the sequences than crude samples including the whole genomic DNA of cattle. After determined the feasibility and accuracy of the experimental workflow with cloned plasmids, the next step to be validate with BLV infected cattle blood DNA.
2) In this manuscript, the authors concluded that the identification of identical BLV proviral sequences as the results of vertical transmission. However, the possibility that dam and daughter horizontally infected by same origin of infected cattle need to be considered. Moreover, during persistent infection, proviral sequences will possibly acquire the mutation by the effect of host immune systems, several APOBEC proteins or incorrect viral RNA polymerase. The authors need to clearly describe the evaluation criteria on vertical transmission and horizontal transmission, for example exact number of SNPs per 7.5Kb, distance on phylogenetic tree, bootstrap value, and so on.
3) The authors demonstrated the accuracy of both PCR techniques and ONT sequencing by the results of matching on taxa location on the tree in Fig 4. However, the two laboratories used different primers with different PCR efficacy and different PCR enzymes with different fidelity. To demonstrate the accuracy by technical replicates, it is necessary to use same primers and same PCR enzymes in two labs.
Minor comments
4) Page 2, Line 75: Previous study (Polat. et al., 2017) has demonstrated that the topology of phylogenetic tree of complete sequences of BLV proviral DNA is similar to that of envelope sequences.
5) Page3 Table1: The description of the reason why primers were set based on AF033818 (isolated in USA) and AP019598.1 (isolated in Japan) are needed.
6) Page7 line165: daughters 5-8?
7) Page9 line209: The random mutation rate and sequence length of six genes need to be considered.
8) Page9 line 207: change “the EPP protein” to “ENV protein” (Page7 line182) or “Env polyprotein (EPP)”
9) Page9 line 204: “dam daughter pair 1 (5SNPs)”, Is this means five single nucleotide polymorphisms (SNPs) with 4 amino acid changes (Table4, blue boxes of dam daughter pair 1)?
Author Response
Reviewer 1
Point 1: The authors mainly relied on the plasmid-based analysis to validate ONT sequencing. The aliquot of samples used in the validation only contains cloned plasmid DNA, which is much easier to analyze the sequences than crude samples including the whole genomic DNA of cattle. After determined the feasibility and accuracy of the experimental workflow with cloned plasmids, the next step to be validate with BLV infected cattle blood DNA.
Response 1: We thank the reviewer for acknowledging the different complexity of template DNA used for PCR-targeted ONT sequencing for the validation of SNP call rate accuracy. Both TOPO-amplified and gDNA-amplified PCR products were sequenced and no genomic DNA was added to the ONT flow cell. This overcomes the aspect of attempting to interpret crude DNA sequencing for this analysis as both PCR amplicon species sequenced were analogous. We felt this approach offered the highest degree of accuracy due to the Sanger sequencing verified inserts within a previously established vector backbone of the TOPO 2.1. The approach employed allowed for more expeditious validation of the method. However, for future studies, sanger verified endogenous ‘housekeeping’ genetic loci can be used to internally validate the integrity of the sequencing approach to ensure the accuracy of BLV genome sequencing by ONT.
Point 2: In this manuscript, the authors concluded that the identification of identical BLV proviral sequences as the results of vertical transmission. However, the possibility that dam and daughter horizontally infected by same origin of infected cattle need to be considered. Moreover, during persistent infection, proviral sequences will possibly acquire the mutation by the effect of host immune systems, several APOBEC proteins or incorrect viral RNA polymerase. The authors need to clearly describe the evaluation criteria on vertical transmission and horizontal transmission, for example exact number of SNPs per 7.5Kb, distance on phylogenetic tree, bootstrap value, and so on.[LP1]
Response 2: We agree with the reviewer, animals found to have identical BLV proviral sequences may have both been infected from a common animal. The main focus of this study was to validate the approach of long-range PCR followed by ONT sequencing for the future potential of using this validated tool for such interpretation. We have made changes to the result section 2.3 and reserved speculation of routes of transmission to the discussion section. We have also included a sentence describing other possible explanations for the identical sequences seen in dams and daughters as you had mentioned.
Point 3: The authors demonstrated the accuracy of both PCR techniques and ONT sequencing by the results of matching on taxa location on the tree in Fig 4. However, the two laboratories used different primers with different PCR efficacy and different PCR enzymes with different fidelity. To demonstrate the accuracy by technical replicates, it is necessary to use same primers and same PCR enzymes in two labs.
Response 3: We thank the reviewer for acknowledging the intricacies and implications of long-range PCR targeted-sequencing using different primers and PCR chemistries. We also considered the recommended approach. However, we felt that it was a stronger and more convincing approach to use independently developed primers and differing PCR chemistries on the same gDNA samples allowing for a more robust evaluation of ONT workflow and resulting accuracy. Given our findings (fig 3-4), we are convinced that the approach described in this manuscript is suitable for genotyping large genetic elements independent of polymerase specific effects.
R1 Minor Comments
Point 4: Page 2, Line 75: Previous study (Polat. et al., 2017) has demonstrated that the topology of phylogenetic tree of complete sequences of BLV proviral DNA is similar to that of envelope sequences.[LP1]
Response 4: We thank the reviewer for point this point of clarity within the introduction. We have revised this section to point out that no single-molecule, contiguous sequencing has been published and that this type of phylogenetic analysis has been performed in the USA.
Point 5: Page3 Table1: The description of the reason why primers were set based on AF033818 (isolated in USA) and AP019598.1 (isolated in Japan) are needed.
Response 5: We have worked to describe the approach, that both labs had similar objectives and methods were established prior to the collaboration taking place and that this approach chosen demonstrates independent novel approaches resulting in very similar outputs. Table 1 was shown to describe the different primer compositions and locations, respective to the references used.
Point 6: Page7 line165: daughters 5-8?
Response 6: Thank you for catching this. Changes have been made
Point 7: Page9 line209: The random mutation rate and sequence length of seven genes need to be considered.
Response 7: This is a limitation of this study. We focused on SNPs which resulted in amino acid substitutions instead of nucleotide diversity. This pilot study does not have sufficient depth to assess random mutation rate, but we do plan to assess this within herds and across herds in the future with several times the number of animals studied in the manuscript herein.
Point 8: Page9 line 207: change “the EPP protein” to “ENV protein” (Page7 line182) or “Env polyprotein (EPP)”
Response 8: Changes to protein name have been corrected
Point 9: Page 9 line 204: “dam daughter pair 1 (5SNPs)”, Is this means five single nucleotide polymorphisms (SNPs) with 4 amino acid changes (Table4, blue boxes of dam daughter pair 1)?
Response 9: We agree that there was confusion regarding the terminology here. The “5 SNPs” was describing AA substitution and not nucleotide changes so the language has been altered accordingly. Also, the 5 was a mistake and should have been 4 to describe the number of AA changes between Daughter and Dam pair 1. Thank you for pointing out these inaccuracies in the text.
Reviewer 2
R2 Minor Comments
Point 1: The last paragraph of the introduction seems to be repeating the previous paragraphs in a slightly different way and could be removed or reworded to avoid redundancy.
Response 1: We agree. The introduction has been consolidated for greater clarity and redundant information has been removed.
Point 2: There have been a few publications using whole genome sequencing to phylogenetically classify BLV, I think these could be referred to better in the introduction and / or text. Currently my reading of the text gives the impression that this has never been done before.
Response 2: Agreed. Please refer to Reviewer 1 Response 4. The text has been edited to include previous whole-genome sequencing and now highlights the novelty of using single-molecule ONT sequencing.
Point 3:Table 1 and table 2 have the same legend.
Response 3: We thank the reviewer for bringing this to our attention. The correct table legend has been added to table 2.
Point 4: I don't think the dot plot serves much purpose - it's not a good way to visually represent information. I would prefer to see an alignment to be honest - perhaps in the supplementary materials?
Response 4: We agree with the reviewer, however, the alignment to replace the dot plot is around 5000bp long. The dot plots allow the 100% identity to be easily visualized as well as exemplifies the length of reads even more so than the identity. We chose this option to highlight the identity over the entire amplicon rather than publishing a multiple sequence alignment which would be too large.
Point 5: I like that you have used different genes to compare Sanger with Nanopore because that is a constant concern with nanopore, however, in a way, the best comparison would have been to target different pieces of the virus genome by Sanger and Nanopore - would this be possible? I don't see this as a make or break experiment that must be done, but it would improve the manuscript if it was done, I think.
Response 5: We agree and appreciate the reviewer acknowledging the potential to Sanger verify a section of the viral genome along with the bacterial inserts used in this study. Please refer to Reviewer response 1.
Point 6: Having a comparison of two laboratories doing effectively the same thing is a lovely aspect of this study. Is it a concern however, that there were differences identified? I think this could be discussed in more detail in the discussion and some possible ideas around improving the robustness of the protocols to increase consistency would be helpful.
Response 6: We agree with the reviewer and thank them for highlighting this concern. There is likely to be differences when comparing two PCRs using different primers, chemistries, and machines. However, despite these differences, the majority of the resulting consensus sequences had 100% identity to each other which confirms the accuracy of the sequencing workflow we developed to overcome the variation between amplicons generated differently. Please refer to Reviewer response 3.
Reviewer 2 Report
Thank you for this manuscript, I enjoyed reading it. it is nicely laid out and gives nice detail.
I had a few minor comments:
1. The last paragraph of the introduction seems to be repeating the previous paragraphs in a slightly different way and could be removed or reworded to avoid redundancy.
2. There have been a few publications using whole genome sequencing to phylogenetically classify BLV, I think these could be referred to better in the introduction and / or text. Currently my reading of the text gives the impression that this has never been done before.
3. Table 1 and table 2 have the same legend.
4. I don't think the dot plot serves much purpose - it's not a good way to visually represent information. I would prefer to see an alignment to be honest - perhaps in the supplementary materials?
5. I like that you have used different genes to compare Sanger with Nanopore because that is a constant concern with nanopore, however, in a way, the best comparison would have been to target different pieces of the virus genome by Sanger and Nanopore - would this be possible? I don't see this as a make or break experiment that must be done, but it would improve the manuscript if it was done, I think.
6. Having a comparison of two laboratories doing effectively the same thing is a lovely aspect of this study. Is it a concern however, that there were differences identified? I think this could be discussed in more detail in the discussion and some possible ideas around improving the robustness of the protocols to increase consistency would be helpful.
Author Response

(The authors gave the same response as above.)
